# Clearing the Fog: A Scoping Literature Review on the Ethical Issues Surrounding Artificial Intelligence-Based Medical Devices

**DOI:** 10.3390/jpm14050443

**Published:** 2024-04-23

**Authors:** Alessia Maccaro, Katy Stokes, Laura Statham, Lucas He, Arthur Williams, Leandro Pecchia, Davide Piaggio

**Affiliations:** 1Applied Biomedical Signal Processing Intelligent eHealth Lab, School of Engineering, University of Warwick, Coventry CV4 7AL, UK; alessia.maccaro@warwick.ac.uk (A.M.); katy.stokes@warwick.ac.uk (K.S.); laura.statham@warwick.ac.uk (L.S.); lucas.he.23@ucl.ac.uk (L.H.); arthur.austin.williams@hsbc.com (A.W.); leandro.pecchia@unicampus.it (L.P.); 2Warwick Medical School, University of Warwick, Coventry CV4 7AL, UK; 3Faculty of Engineering, Imperial College, London SW7 1AY, UK; 4Intelligent Technologies for Health and Well-Being: Sustainable Design, Management and Evaluation, Faculty of Engineering, Università Campus Bio-Medico Roma, Via Alvaro del Portillo, 21, 00128 Rome, Italy

**Keywords:** artificial intelligence, machine learning, healthcare, medical devices, regulatory affairs, ethics

## Abstract

The use of AI in healthcare has sparked much debate among philosophers, ethicists, regulators and policymakers who raised concerns about the implications of such technologies. The presented scoping review captures the progression of the ethical and legal debate and the proposed ethical frameworks available concerning the use of AI-based medical technologies, capturing key themes across a wide range of medical contexts. The ethical dimensions are synthesised in order to produce a coherent ethical framework for AI-based medical technologies, highlighting how transparency, accountability, confidentiality, autonomy, trust and fairness are the top six recurrent ethical issues. The literature also highlighted how it is essential to increase ethical awareness through interdisciplinary research, such that researchers, AI developers and regulators have the necessary education/competence or networks and tools to ensure proper consideration of ethical matters in the conception and design of new AI technologies and their norms. Interdisciplinarity throughout research, regulation and implementation will help ensure AI-based medical devices are ethical, clinically effective and safe. Achieving these goals will facilitate successful translation of AI into healthcare systems, which currently is lagging behind other sectors, to ensure timely achievement of health benefits to patients and the public.

## 1. Introduction

The medical device market, including medical software, is one of the fastest growing sectors in the world, at much higher rates than adjacent sectors, i.e., biotechnology and pharmaceuticals. Eight thousand new medical device patents were granted in Europe during 2021 alone [1]. Artificial intelligence (AI) in particular, with its capabilities and promises, is among the main drivers of the Fourth Industrial Revolution, which is set to transform all sectors, including that of healthcare (Health 4.0). Health 4.0 relies on Industry 4.0 concepts and principles, characterised by increased interconnectivity between cyber and physical elements and solutions based on disruptive information and communication technologies (ICTs) (e.g., big data and the Internet of Things), to shift from hospital-centred to patient-centred organisations and services [2,3,4]. Developments in medical technology are occurring faster than ever before, with Healthcare 5.0 already being the new desirable frontier in terms of smart disease control and detection, smart health monitoring and management and virtual care, relying on emerging digital technologies (e.g., nanotechnology, 5G technologies, robotics, IoT, big data, AI, cloud computing, etc.). Healthcare 5.0, with the introduction of intelligent sensors, is set to overcome the limits of Healthcare 4.0, i.e., the lack of emotive recognition [5,6].

AI is a broad term referring to the capacity of computers to mimic human intelligence. Currently, the majority of healthcare applications of AI relate to machine learning for specific tasks, known as artificial narrow intelligence. There is also interest and debate on the future use of artificial general intelligence (able to reason, argue and problem solve) and artificial superintelligence (cognitive capacity greater than that of humanity), which are at a much earlier stage of research and development [7]. AI is increasingly infiltrating healthcare, unlocking novelties that in some cases were difficult to imagine. AI has shown huge potential across many areas including improved diagnosis and disease monitoring (e.g., using wearable sensors) and improved operational services (e.g., forecasting of pharmaceutical needs) [8]. This is reflected in the AI-in-healthcare market that was valued at USD 8.23 billion in 2020 and is set to reach USD 194.4 billion by 2030 [9]. The number of approvals for AI-enabled medical devices by the Food and Drug Administration follows this rapid growth: in the last seven years there was a surge from less than ten approvals yearly to approximately 100 [10]. Among the other benefits, a reduction in healthcare costs is envisaged. For example, Bohr et al. claim that US annual healthcare costs can be cut by USD 150 billion in 2026 with AI applications [11].

This wave of technological progress requires fast-responsive regulations and guidelines, to safeguard patients and users. Whilst there are several global initiatives for AI regulation in development, these are still in progress. For example, currently, AI is not specifically regulated in the UK, though the UK government recently launched a public consultation on what an AI regulation might look like. In March 2023, the Office for AI of the UK Department for Science, Innovation and Technology, published a policy paper highlighting the envisioned UK regulatory approach with a focus on mitigating risks while fostering innovation, pinpointing the design and publication of an AI Regulation Roadmap as one of the paramount future steps [12]. In terms of UK regulations for AI-based medical devices, there is nothing published yet, as the novel UK medical device regulations are expected to be laid in Parliament only by mid-2025. For this reason, referring to one of the most recent regulations for medical devices, i.e., the European ones, is more proper for this article’s purposes.

In Europe, AI systems that qualify as medical devices are regulated as such by the medical device Regulations (EU2017/745), which were published in May 2017 and came into effect in 2021 to ensure greater safety and effectiveness of MDs. The MDR introduced a more stringent risk-based classification system for medical devices, leading to increased scrutiny and regulation for higher-risk devices and for devices that were previously not classed as medical devices (e.g., coloured contact lenses). Despite this positive goal, the MDR still face related risks and challenges that cannot be underestimated [13]. While the MDR regulate software with intended medical purpose as a MD, i.e., software as a medical device (SaMD) or medical device software (MDSW), it does not mention, nor specifically regulates, AI-based MDs. In fact, the need to regulate AI-based MDSW first stemmed from the action plan published by the U.S. Food and Drug Administration (FDA). The section “Evolution of AI regulatory frameworks” summarises the main relevant changes and milestones in the AI regulatory landscape [14].

Since the medical device sector is rapidly expanding, especially in terms of AI-based solutions, it becomes imperative to find a proper way of integrating novel frameworks to regulate such devices in the MDR. This should be led by a joint interdisciplinary effort, comprising biomedical engineers, computer scientists, medical doctors, bioethicists and policymakers, who should have the necessary competence, networks and tools to ensure [15,16,17,18,19,20,21,22,23,24,25,26] proper consideration of ethical matters in the conception, design, assessment and regulation of new AI technologies. In fact, this fast-paced evolution also requires the continuous training and education of the relevant aforementioned figures. Globally, there are already some initiatives in this direction, which allow medical students to acquire skills and knowledge on the cutting-edge technologies and devices that they will use in their daily practice [27,28,29,30,31]. Nonetheless, compared to the technical aspects, the ethical components are often overlooked and are not currently included in the biomedical engineering curricula [32].

Currently, the use of AI in healthcare has sparked much debate among philosophers and ethicists who have raised concerns about the fairness, accountability and workforce implications of such technologies. Key values relating to the use of AI in healthcare include human agency and oversight, technical robustness and safety, privacy and data governance, transparency, diversity, nondiscrimination and fairness, societal and environmental well-being and accountability [15].

In order to address this, this paper presents the results of a scoping review with the aim of investigating and “clearing the fog” surrounding the ethical considerations of the use of AI in healthcare.

This specific project is the natural continuation of previous works, in which we have already described how the word “ethics” is currently widely (ab)used in scientific texts, but without real competence, almost as if it were a “humanitarian embellishment” [33,34]. One of the most recent consequences of this is the publication of numerous articles about COVID-19 that were not scientifically sound and were then retracted [33,34]. This is perfectly aligned with our previous work that shows how an “ethics by design” and frugal approach to the design and regulation of medical devices via ad hoc frameworks is key for their safety and effectiveness [35,36,37].

## 2. Evolution of AI Regulatory Frameworks

This section summarises the main documents published by groups of ad hoc expert committees on AI in the latest year relative to the use of AI. These documents vary considerably for countries and regions: e.g., the expert group on AI in the Society of the Organisation for Economic Co-operation and Development (OECD) issued guidelines for the responsible use of AI [16]. The private sector has also developed policies relating to AI (i.e., Google in 2018 [17]). Declarations and recommendations have also been issued by professional associations and nonprofit organisations such as the Association of Computing Machinery (ACM) [18], Access Now and Amnesty International [19].

As an inherently data-driven technology, data protection regulation is highly relevant to AI. In the European Union, the General Data Protection Regulation (GDPR) [20] includes the right to an explanation for automated decision making and the need for data protection impact assessments. There is an ongoing need for consistency in ethical guidance across these regulatory frameworks, that allows stakeholders to shape the ethics of AI in ways that meet their respective priorities [21].

In this landscape, it is recognised that there is a general lack of public understanding and trust in AI. Bodies such as the High-Level Expert Group on Artificial Intelligence (AI HLEG) [38], an independent expert group set up by the European Commission in June 2018, have set out to address this. As a result, AI HLEG published the Ethics Guidelines for Trustworthy Artificial Intelligence in April 2019, outlining guiding principles for trustworthy AI and setting out key requirements to be met by these systems [22]. Based on fundamental rights and ethical principles, these Guidelines list seven key requirements that AI systems should meet in order to be trustworthy, namely human agency and oversight, technical robustness and safety, privacy and data governance, transparency, diversity, nondiscrimination and fairness, societal and environmental well-being and accountability. The document also lists some principles, including human autonomy, the prevention of harm, fairness, explicability and explainability.

On the 17 of July 2020, the same group presented their final Assessment List for Trustworthy Artificial Intelligence (ALTAI), a tool designed to support the development of trustworthy AI, in accordance with the key requirements outlined by the Ethics Guidelines for Trustworthy Artificial Intelligence [23]. In the same year, the European Commission published a White Paper entitled “On Artificial Intelligence—A European approach to excellence and trust” [24]; the paper works on the objectives of “promoting the uptake of AI and of addressing the risks”. A key idea from the paper is to outline the requirements applicable to high-risk AI uses. These requirements (stated in section 5D of the aforementioned paper), aim to increase the trust of humans in the system, by ensuring it is ethical. On the 23 of November 2021, UNESCO’s 193 Member States adopted the “UNESCO Recommendation on the Ethics of Artificial Intelligence”, the first global normative instrument on the ethics of AI, which addresses the concerns about the protection of life during war and peace when using AI and lethal autonomous robots, outlined in Human Rights Council Resolution 47/23 [25].

The latest achievements in this field are quite recent. In December 2023, an agreement was reached between the EU Parliament and the Council on the EU AI Act, the first-ever comprehensive legal framework on AI worldwide, that was proposed by the EU Commission in April 2021 [39]. The President of the EU Commission, Ursula von der Leyen, commented the following: “The AI Act transposes European values to a new era. By focusing regulation on identifiable risks, today’s agreement will foster responsible innovation in Europe. By guaranteeing the safety and fundamental rights of people and businesses, it will support the development, deployment, and take-up of trustworthy AI in the EU. Our AI Act will make a substantial contribution to the development of global rules and principles for human-centric AI”. Furthermore, it is also worth mentioning the World Health Organization (WHO) involvement in this scenario. In 2024, in fact, the WHO published a document titled “Ethics and governance of artificial intelligence for health: Guidance on large multi-modal models”, specific to generative AI applications in healthcare [26].

In this evolving scenario, as AI-based medical devices directly concern human life, health and wellbeing, it is essential that the relevant ethical declaration and principles are respected in all regulatory approaches. In fact, any error caused by the use of AI in healthcare may have severe consequences, either directly, such as failure to diagnose lethal conditions, or more widely, such as by leading to a deskilling of the healthcare workforce. See Appendix A for further details on the relevant existing laws related to the main topics of ethical concerns about AI.

## 3. Methods

### 3.1. Search Strategy

This scoping literature review was conducted according to PRISMA guidelines (Preferred Reporting Items for Systematic Reviews and Meta-Analyses) [40]. The protocol for the review was not registered. The search was run from database inception to April 2022 and updated in September 2023. The three main topics, i.e., ethics, AI, healthcare technology, were then put together with the AND operator, see Table 1.

### 3.2. Study Eligibility

Only scientific articles focusing on ethical challenges/issues arising from medical AI/technology were included. Studies were excluded if the following criteria were met: non-English language, the full text was not freely accessible, grey literature, letter to editors, editorials, commentaries or review articles.

### 3.3. Study Selection

Two authors independently screened the studies by title and abstract while three authors completed full text screening against the inclusion and exclusion criteria, with conflicting decisions being mitigated by an additional reviewer. Relevant data were then extracted using an ad hoc extraction table to facilitate the analysis and narrative synthesis.

### 3.4. Data Extraction

Relevant data were extracted and collected in an ad hoc Excel sheet, organised by author, main ethical issues raised, technological and medical context and key findings.

### 3.5. Data Synthesis

To synthesise the extracted data, a narrative synthesis method was used [41]. For each study, the main ethical issues and their solutions were identified, which were then described in the results, organised by principal themes.

## 4. Results

### 4.1. Study Selection

A total of 2059 studies were returned through the search strategy and 73 of these were included in the final review, considering the fulfilment of our inclusion and exclusion criteria. The study selection process is summarised in Figure 1.

### 4.2. Study Characteristics

The 73 included studies covered a broad spectrum of medical contexts, summarised with the main ethical issues raised and key findings in Appendix A. Briefly, the most frequently addressed areas were as follows: the general use of AI for healthcare (36 studies) [42,43,44,45,46,47,48,49,50,51,52,53,54,55,56,57,58,59,60,61,62,63,64,65,66,67,68,69,70,71,72,73,74,75,76,77], the use of AI in decision support systems (eight studies) [78,79,80,81,82,83,84,85,86,87,88,89,90,91,92,93,94], big data (four studies) [95,96,97,98], robotics (seven studies) [99,100,101,102,103,104,105] and adaptive AI (one study) [106]. The remaining studies addressed the following: rehabilitation [107], medical education [108], monitoring technology for the elderly [109], mental health [110], radiation technology [111], chatbots [112], health apps [113] and healthcare in low- and middle-income countries [114].

Across the 73 selected studies, eight main ethical themes concerning medical technology were identified including the following: transparency, algorithmic bias, confidentiality, fairness, trust, autonomy, accountability and informed consent. For the purpose of this study, the ethical themes are defined in the context of AI, as follows:

Transparency: “the data, system and AI business models should be transparent. Traceability mechanisms can help achieving this. Moreover, AI systems and their decisions should be explained in a manner adapted to the stakeholder concerned. Humans need to be aware that they are interacting with an AI system, and must be informed of the system’s capabilities and limitations” [22].

Algorithmic bias: “the instances when the application of an algorithm compounds existing inequities in socioeconomic status, race, ethnic background, religion, gender, disability or sexual orientation to amplify them and adversely impact inequities in health systems” [115].

Confidentiality: “the responsibility of those entrusted with those data to maintain privacy” [116].

Fairness: “a commitment to ensuring equal and just distribution of both benefits and costs, and ensuring that individuals and groups are free from unfair bias, discrimination and stigmatisation” [22]

Trust: “a set of specific beliefs dealing with benevolence, competence, integrity, and predictability; the willingness of one party to depend on another in a risky situation (trusting intention); or the combination of these elements” [22].

Autonomy: “The right of an individual to make his or her own choice” [117].

Accountability: “Mechanisms should be put in place to ensure responsibility and accountability for AI systems and their outcomes. Auditability, which enables the assessment of algorithms, data and design processes plays a key role therein, especially in critical applications. Moreover, adequate an accessible redress should be ensured” [22].

Informed consent: “Consent of the data subject means any freely given, specific, informed and unambiguous indication of the data subject’s wishes by which he or she, by a statement or by a clear affirmative action, signifies agreement to the processing of personal data relating to him or her” [20].

The frequencies of reporting of the ethical themes are displayed in Table 2, along with the proposed solutions. The selected studies focused on varied technological contexts, displayed in Figure 2. A summary of the medical contexts covered in the included studies is provided in Appendix A.

### 4.3. Transparency

The most frequently addressed ethical theme was transparency (n = 40) [44,46,47,49,55,56,57,58,59,60,61,62,64,66,69,70,75,76,78,79,80,81,82,85,86,88,89,90,92,94,95,96,99,100,101,107,110,111,113,114], concerning a wide range of medical contexts but focusing mainly in the area of clinical decision support. While the majority of studies reported a narrative evaluation of ethical issues, seven qualitative studies employed surveys or focus groups that identified transparency as an ethical priority by either healthcare professionals [44,78], experts in AI implementation [107], a combination of AI experts and citizens [65] or a combination of designers, clinician-users, administrators and patients, [69,70,89]. For example, de Boer et al. compared concerns of healthcare professionals regarding clinical decision support systems, describing theories of ‘technomoral change’ and ‘technological mediation’ as a means to reflect and identify key concerns of machine learning in healthcare [78].

Cagliero et al. performed a case study analysis and collected viewpoints from various stakeholders, noting that while designers appeared focused on the importance of algorithm validation, clinicians and patients had a desire to understand (at least at a certain level) how the AI system works [89]. This demonstrates a so-called ‘values-collision’, highlighting that it should not be taken for granted that all stakeholders assume the same viewpoint of what is required for transparency. This can also be seen to reflect two sides of the discussion of transparency, on the one hand, there can be seen a need for rigorous validation which is communicated well, on the other hand, regardless of the validation, the ability to understand the AI processes is inherently important to people. This was a key discussion point which was raised in depth in several explorations of transparency. Indeed, a critical and somewhat unique challenge raised by the nature of AI-driven technologies is the extent to which it is possible to understand how the underlying AI system works or reaches its output [49,60,62,64,66,70,85,86,88,92]. Many studies called for this ‘explicability’ or ‘explainability’ as a key principle to be met for AI health technologies [60,64], while others emphasised the need for contextual explainability [88]. Adams et al. provided arguments for the inclusion of explainability as a principle of bioethics, negating challenges from others that medical decision making by people themselves is inherently not transparent [64]. It is not surprising, therefore, given the range of arguments, that it was noted by several authors that for the final AI-based device to be considered transparent and acceptable for use, stakeholders such as patients and the public should be involved early in the development process [58,66,80,89,90,95,111]. This was supported by several studies highlighting the need for a multidisciplinary approach to the issue of transparency and ethics more widely (specifically, ethics training or integrating ethicists) [44,47,59,69,75,100,101]. In terms of barriers, AI development experts identified a lack of transparency to be seen as a limit to positive relationships between vulnerable patient groups (such as patients of older generations) and clinicians [107].

More broadly, several studies noted that the public should have access to transparent information regarding the use of AI in healthcare [61,66,69,76], with some pinpointing that governments should play a role in raising awareness, especially of prospective vulnerable populations, such that information provided regarding AI-based devices can be well understood [96]. Further, to facilitate transparency, AI standardised ethical frameworks and clear regulations are required, with some studies calling to enforce their use with legislation [46,56,57,75,82,94,107,110,114].

### 4.4. Accountability

Accountability emerged as a theme from 25 studies, particularly from studies concerning applications of large language models/chatbots and robotics [42,46,55,56,60,61,69,71,74,75,78,79,80,81,84,88,97,100,101,102,103,104,110,111,112].

Several authors highlighted the importance of accountability at the patient care level, underpinning trust in the patient–clinician relationship, which may be changed or challenged by the use of chatbots and decision support systems [79,81,101,102,112]. Studies called for clear models for cases of investigation of medical accidents or incidents involving the use of AI [55,110]; one study emphasised this as a necessity in order to truly prepare healthcare systems for the use of AI [104]. In this way, legal accountability must be made clear for applications of AI decisions or decision support [74]. Implementation of AI needs to also be supported by ethical design and implementation frameworks or guidelines, for which designers are accountable to meet [56,61,71,97,111]. In some cases, authors advocated for ensuring medical AI is always ‘supervised’ by a healthcare professional, who ultimately has accountability for the technology [61].

As with transparency, multidisciplinarity (specifically, training and integrating ethicists) was raised as essential in ensuring acceptable levels of accountability in health-related decisions [42,74,78,80,84,100,101].

### 4.5. Confidentiality

A total of 33 of the included papers examined the challenges and considerations surrounding the use of AI, particularly in healthcare, and the imperative of safeguarding confidentiality in the digital age [42,43,44,46,47,50,53,54,58,59,60,61,66,67,68,71,72,74,75,76,84,90,93,94,96,98,102,104,105,109,110,111,113].

Broadly, these papers address the complexity of maintaining confidentiality in an era where AI technologies are increasingly integrated into healthcare systems. Key themes that emerged include the tension between technological advancement and ethical constraints, the impact of AI on patient privacy and data security, and the moral obligations of AI developers and users towards ensuring the confidentiality of sensitive information.

A significant number of papers focus on the use of AI in healthcare, particularly concerning patient data privacy and security. This group explores the challenges and ethical considerations in safeguarding patient information in the context of AI-driven medical practices and health apps [43,44,46,47,53,60,61,68,71,84,90,102,105,113]. Technological and practical challenges in ensuring confidentiality was also addressed [53,61,104]. This included discussions on data encryption, secure data handling practices and the implementation of robust security measures to prevent data breaches.

In summary, the collective insights from these papers underscore the critical need for robust ethical frameworks and security measures in AI applications, particularly in healthcare. They highlighted the imperative of balancing technological innovation with the ethical responsibility to protect confidentiality and maintain the integrity of patient data. The papers suggested a multidisciplinary approach, involving stakeholders at various levels, to address these challenges effectively. There was a consensus on the need for ongoing dialogue, policy development, and ethical guidelines to navigate the complex landscape of AI and confidentiality.

### 4.6. Autonomy

The theme of autonomy was mentioned in 26 of our inclusions, and was reviewed on various aspects, from philosophical and ethical foundations to practical implications in healthcare and other domains [43,46,48,49,50,56,59,61,64,69,71,74,76,79,84,85,87,88,91,99,100,105,109,113]. The theme intricately linked with issues such as decision making, control, and the human-centric approach to AI development and implementation.

Sevesson et al. discussed the impact of AI on human dignity and autonomy, emphasising the need to maintain the uniqueness and intrinsic value of humanity. Kuhler M et al. explored paternalism in health apps, including fitness and wellbeing applications, and its implications for autonomy, particularly in AI-driven tools [113]. Braun M et al. delved into decision-making complexities in AI contexts and introduced the concept of ‘Meaningful human control’ as a framework to ensure autonomy in AI systems [79]. Compliance with universal standards in AI, particularly stressing the importance of maintaining autonomy in the face of technological advancements, was proposed by Arima et al. [46]. Similarly, Guan et al. addressed the application of AI in various sectors, advocating for specific guidelines to ensure autonomy, especially in frontier technologies [48].

The central tenet of these papers called for the imperative to preserve and respect human autonomy in the face of rapidly advancing AI technologies. The authors collectively emphasised that AI should be developed and implemented in ways that enhance human decision making and independence, rather than undermining it.

### 4.7. Algorithmic Bias in AI-Based Medical Devices

Algorithmic bias refers to the tendency of AI systems to exhibit systematic errors that disproportionately affect certain patient groups, directly influencing patient treatment and the efficacy of medical interventions. This bias can emerge from various sources, including biased training data or flawed algorithm design, leading to unequal treatment outcomes. Studies from Kerasidou [114] and McLennan et al. [42] highlighted the ethical need to address these biases in smart healthcare systems. They argued for the development of AI systems that prioritize equity and fairness, ensuring that these technologies serve as reliable and unbiased tools in medical diagnostics and treatment. Zhang and Zhang emphasised the critical role of transparency in AI systems, advocating for measures that prevent the deepening of existing healthcare disparities through biased algorithms [61]. Similarly, Liu et al. discussed the ethical challenges posed by digital therapeutics, including AI-driven devices, stressing the importance of considering diverse patient populations during development to mitigate bias [98]. Hallowell et al. explored the ethical considerations in the design of medical IoT devices, also emphasising the need for inclusive and fair algorithms that cater to diverse patient needs [87].

### 4.8. Informed Consent in the Era of AI

Informed consent in AI-enhanced medical care has become increasingly complex. The concept extends beyond the traditional model of patient–physician interaction, incorporating the understanding of AI-driven processes that influence patient care. Lorenzini et al. [88] and Astromskė et al. [82] addressed the complexities involved in obtaining informed consent when medical decision making is augmented with machine learning, emphasising the need for clarity in communication. Leimanis and Palkova [108] and Parviainen and Rantala [112] further discussed the principle of patient autonomy in this context, highlighting the right of patients to make informed decisions about their care, particularly when influenced by advanced medical technologies. Astromskė et al. [82] delved into the practical challenges of ensuring informed consent in the context of AI-driven medical consultations, suggesting strategies to enhance patient understanding and autonomy. Ho discussed the ethical considerations in using AI for elderly care, particularly focusing on the need for clear consent processes tailored to this demographic [109].

### 4.9. Intersection of Algorithmic Bias and Informed Consent

The intersection of algorithmic bias and informed consent presents unique challenges in AI-based medical care. Biased algorithms can obscure decision-making processes, consequently affecting the ability of patients to provide informed consent. Transparent AI systems, as advocated in [106,113], are essential to ensure that patients understand how biases in AI might impact their healthcare. Education and awareness, as highlighted in [95,98], play a vital role in enabling informed patient–provider discussions about the role of AI in healthcare. The ethical deployment of AI in medical devices necessitates a comprehensive understanding and mitigation of algorithmic bias and informed consent challenges. Collaborative efforts involving technology design, patient education, regulatory frameworks, and ethical considerations are paramount. This collaborative approach ensures the development of equitable, transparent and patient-centred healthcare solutions. It is through this integrated perspective that AI in healthcare can be effectively navigated, ensuring that its benefits are maximized while minimizing potential harms and ethical complexities.

### 4.10. Trust

Trust was discussed as a theme in 34 of the included studies [42,50,52,54,58,59,60,61,62,63,66,67,68,72,73,76,77,79,80,81,82,86,87,89,90,93,94,101,102,105,107,110,112,113], with most focusing on clinical decision support systems, chatbots and robots. Most of these studies were articles or qualitative analyses. The main concern raised within this theme was the impact of untrustworthy AI on clinician–patient relationships. Several studies described how building a reliable doctor–patient relationship relies upon the transparency of the AI device [81,107], as previously discussed. Interviewees of one qualitative study described how the perceived reliability and trustworthiness of AI technology relies upon validating its results over time, and bias is a significant problem that may impair this [87]. Arnold also described how AI devices may erode trust if doctors do not have the autonomy or control of these devices [50]. Braun et al. echoed this, suggesting ‘meaningful human control’ must be developed as a concept to stand as a framework for AI development, especially in healthcare where decisions are critical [79].

Medical chatbots were discussed as a mode for increasing rationality but also leading to automation, which may lead to incompleteness, and, therefore, a loss of trust [112]. De Togni described how human and machine relationships are uncertain in comparison, and there is a need to rematerialize the boundaries between humans and machines [70]. Other recommendations given to improve trustworthiness included multidisciplinary collaboration, for example engaging with both clinicians, machine learning experts and computer program designers [58,98], more precise regulation [60] and specific guidelines for frontier AI fields [48].

### 4.11. Fairness

A total of 25 studies concerned the topic of fairness, covering a range of contexts [44,45,48,54,56,58,59,66,67,68,69,71,73,80,81,83,85,89,95,98,99,101,103,106,114]. This theme largely discussed justice and resource allocation of AI technology. Pasricha explained how most vulnerable patients do not have access to AI-based healthcare devices, and that AI should be designed to promote fairness [98]. Kerasidou specifically discussed the ethical issues affecting health AI in low- or middle-income countries (LMICs), concluding that further international regulation is required to ensure fair and appropriate AI [114]. This was echoed by others indicating that a revision of guidelines is necessary to ensure fair medical AI technology [56]. Another suggestion was ethicist involvement with AI technology development, with the view that this may improve the chance that AI is fair and unbiased [98].

## 5. Discussion

Overall, the ethical considerations surrounding AI are complex and multifaceted and will continue to evolve as the technology itself advances, although it seems that traditional issues are not yet fully overcome, since they are still a matter of consideration and concern. There is an ongoing need to assess the ethical issues and proposed solutions and to identify gaps and best routes for progress. In particular, common concerns include the following:The lack of transparency in relation to data collection, use of personal data, explainability of AI and its effects on the relationship between the users and the service providers;The challenge of identifying who is responsible for medical AI technology. As AI systems become increasingly advanced and autonomous, there are questions about the level of agency and control that should be afforded to them and about how to ensure that this technology acts in the best interests of human beings;The pervasiveness, invasiveness and intrusiveness of technology that is difficult for the users to understand and therefore challenges the process of obtaining a fully informed consent;The lack of the establishment of a trust framework that ensures the protection/security of shared personal data, enhanced privacy and usable security countermeasures on the personal and sensitive data interchange among IoT systems;The difficulty of creating fair/equitable technology without algorithmic bias;The difficulty of respecting autonomy, privacy and confidentiality, particularly when third parties may have a strong interest in getting access to electronically recorded and stored personal data.

Starting from the aforementioned AI HLEG (EU Commission) Ethics Guidelines for Trustworthy Artificial Intelligence and its four principles, namely respect for human autonomy, prevention of harm, fairness and explicability, it can be noted that, upon closer inspection, they are comparable to the classic principles of bioethics, namely beneficence, nonmaleficence, autonomy and justice. The latter are considered the framework for ethics and AI by Floridi et al., who further adds “explicability, understood as incorporating both intelligibility and accountability”. Autonomy clearly features both lists [118]. Prevention of harm could be seen as parallel to nonmaleficence (i.e., to avoid bias and respect security and privacy). Fairness includes beneficence and justice, not only relative to the individual but to society as well. Findings from this scoping review strongly support the proposition of Floridi et al. to include explainability as a principle of modern bioethics.

The topic of explicability/explainability is also addressed by the AI HLEG document and is related to the ethical theme of transparency, which was addressed in over half of all the studies included in this review. The transparency of AI may also be seen to underpin other ethical concerns including trust, fairness and accountability. In particular, the appropriate selection and use of medical devices relies on an understanding of how they work, which is key to mitigating any possible risks or biases. However, in some cases, it could be challenging or impossible to determine how an AI system reaches an output (e.g., black boxes) and this is well interwoven with the concept of ‘explainability’ of AI, referring to the level of understanding in the way a system reaches its output. The most extreme case is the so-called ‘black box’ systems, where no information is available on how the output is reached. Increasing the explainability of AI algorithms is an active research field and there is a growing number of methods aiming to offer insight as to how AI predictions are reached [119]. However, significant debate remains as to whether it is ever appropriate to deploy algorithms which are unexplainable in healthcare settings. The question of whether (or to what degree) AI must be explainable, and to who, is complex. Poor communication between stakeholders has been identified in previous literature as a limiting factor in the successful development of AI health technologies, with calls for increased representation of diverse ethnic socioeconomic and demographic groups and promotion of open science approaches to prevent algorithmic bias from occurring. Involving interdisciplinary and cross-sector stakeholders (including healthcare professionals, patients, carers and the public) in the design and deployment of AI will help to ensure the technologies are designed with transparency, that they meet clinical needs and that they are ultimately acceptable to users [120,121].

Transparency also relates to autonomy and consent; if a clinician cannot describe the details involved in the AI’s decision-making process, the relevant information may not be communicated to a patient effectively, preventing fully informed consent from taking place. Also, accountability is noteworthy; who can be held responsible for decisions made via clinical decision support systems when the developers cannot explain the decision-making process that has occurred? Leimanis et al., therefore, suggested that AI systems cannot yet be the primary decision maker, rather they should act only as an assistant to clinicians [108].

As demonstrated by the findings of this review, a frequent theme in the debate on ethics, AI and the IoT entails issues related to the sharing and protection of personal data. It has been argued that one key characteristic of the use of the “things” in the IoT is that the collection of information is passive and constantly ongoing, making it difficult for users to control the sharing and use of data. Andrejevic and Burdon described this phenomenon as the “sensor society”, where sensor driven data collection takes place in a complex system, where the collection and analysis target a pattern of data rather than the individual persons and where processes of data collection and analysis are opaque. As a consequence, it is difficult for an individual to anticipate how their individual data will be used [122]. The above discussion highlights the way in which the evolution and roll out of IoT applications is taking place against the backdrop of discussions around trust, transparency, privacy and security.

Health-related data are considered personal and classed as sensitive information throughout the lifecycle (acquisition, storage, transfer and destruction). Due to the sensitivity of the data and the potential consequences for the users, human control over algorithms and decision-making systems is paramount for these applications. For example, as noted in projects related to the IoT and Active and Healthy Ageing (EU Large-Scale Pilot, GATEKEEPER [123]), while the continuous monitoring of personal health data can be very beneficial to improve and personalise treatment, some may worry about ethical issues like constant surveillance and lack of control over the data collected, hindering autonomy and confidentiality. Ho (2020) described how monitoring technology for older adults may be effective in reducing caregivers’ burden and improve the quality of care but may be viewed as an invasion of privacy and can affect family dynamics [109]. This situation is also complicated in cases where patients, for example older people with cognitive impairments, may not be in a position to participate in the decision-making process around privacy settings, but can be supported by either health information counsellors or some AI-based tools (e.g., assistive technologies).

Hence, an urgent need has emerged for a universal (recognised by law) ethical framework that can support all the individuals involved with the use of AI in healthcare. For example, in the medical field, it will assist medical professionals, carers and other health service providers in meeting their moral responsibilities in providing healthcare and management. Likewise, users will be empowered and protected from potential exploitation and harm via the AI technology. By creating and adopting an ethical framework and guidelines, developers could demonstrate a serious commitment to meeting their legal and moral responsibilities to users, care providers and other stakeholders. Furthermore, this may prevent many foreseeable ethical problems in the design and roll out of IoT devices and protocols, for which developers would be legally or morally liable. In ongoing discussions on forming an ethical framework for AI and IoT, trust is a recurring theme. All stakeholders involved in the development, deployment and use of AI and IoT applications need to be ensured that the systems demonstrate their trustworthiness from social, technical and legal perspectives.

In accordance with this principle, as seen in the results of this review, the debate proposes some solutions to develop a framework of ethical guidelines on AI in healthcare. In primis, a potential solution includes the consideration of a multidisciplinary approach [44,85,100,107], or more specifically involving experts from ethics [47,80,111], bioethics [54] and policy [84], encouraging the involvement of the stakeholders [58] and their communication [80]. Multidisciplinarity is intended not only at the theoretical debate level, but also practically, for example, involving physicians in the designs of AI-based medical technology [55,58], along with bioethics and policy experts [54,84] and other stakeholders [58,80]. Other authors referred to embedded ethics [42] as a means of integrating ethics in technology design, development and deployment to minimise risks and fears. For example, Smith proposed an integrated life-cycle approach to AI, integrating ethics throughout transparency, replicability and effectiveness (TREE) [42,111].

Another important point is the standardisation of regulatory frameworks at the international level [46,48,114], in particular offering better guidance for low- and middle-income countries [83]. The main debate considers the choice between improving the existing ethical–legal solutions [46,56,57,81,95,96,102] or proposing new ethical–political approaches and policy decisions [43,112]. In relation to this, it is noteworthy to mention that certain basic ethical principles are indisputable. Therefore, when updating existing guidelines with the latest technological advancements, existing frameworks cannot be disregarded.

Finally, the improvement of training and education on technology for professionals [47,50,55,82,99,107] and the general public [96,107] is paramount. It is essential to not only create cross-sectoral expertise encouraging basic ethical training at schools and universities, but also on the basic elements of technologies. This does not mean that professionals in a field should be experts in all the relevant disciplines. Rather, this basic multidisciplinary knowledge is key to promoting and facilitating the communication on common topics among experts from different disciplines. Creating multidisciplinary teams helps constructive dialogue and prepares citizens for technological advancement without unnecessary fears, but with a full sense of responsibility. In light of this, some authors referred to “health information counsellors” [57,99], who can support patient autonomy regarding healthcare decisions. It is essential to reflect on figures such as ethic counsellors or the ethical committees in research and clinical practice, which are aimed at supporting patients and medical staff with ethical queries and technologies.

In light of this, the authors of this manuscript believe that it is neither necessary nor useful to rethink the basic principles of ethics in order to propose a framework that responds to the new needs emerging from the use of AI in medicine. However, they believe that a specific, context-aware and internationally harmonised approach to the regulation of AI for medical applications is required urgently to “clear the fog” around this topic. Such an approach could be built starting from the principles listed above (i.e., respect for human autonomy, prevention of harm, fairness and explainability or the parallel bioethical ones, i.e., autonomy, nonmaleficence, beneficence and justice with the addition of explainability).

Many of the issues raised here exist more widely in the regulation of medical devices, as some of the authors of this paper have highlighted in previous work [37]. On a similar thread, some of the authors of this project have already proposed the need for frugal regulations for medical devices, declaring that the current regulatory frameworks for medical devices are not aware of peculiar contexts or responsive to their specific needs [37].

All in all, this regulation for the use of AI in the medical field will only be possible through the combination of solutions: defining a unique ethical–legal framework involving multidisciplinary teams and intercultural and international perspectives, involving stakeholders and the public through education in ethics and technology as well as the consultation in the development of guidelines and technology.

## 6. Conclusions

This paper presents the results of a scoping literature review on ethics and AI-based medical technology. The objectives of this review were as follows:Clarifying the ethical debate on AI-based solutions and identifying key issues;Fostering the ethical competence of biomedical engineering students, who are coauthors of this paper, introducing them to interdisciplinarity in research as a good practice;Enriching our already existing framework with the need for considerations of ethical–legal aspects of AI-based medical device solutions, awareness of the existing debates and an innovative and interdisciplinary approach. Such a framework could support AI-based medical device design and regulations at an international level.

The ethics of AI is a complex and multifaceted topic that encompasses a wide range of recurring issues (for example, transparency, accountability, confidentiality, autonomy, trust and fairness), which are not yet addressed by a single and binding legal reference at the international level. For this, the authors of this paper propose several solutions (interdisciplinarity, legal strength and citizenship involvement/education) in order to reinforce the theories presented in their legal–ethical framework. This tool, intended to support the development of future health technologies, is adaptable and versatile and in continuous refinement.

In conclusion, this work is a step forward in understanding the ethical issues raised by novel AI-based medical technologies and what guidance is required to face these challenges and prevent patient/user’s harm. Although this work is focused on the ethical debate on AI-based medical technologies, it sits well in the wider context of that relative to ethics and technology, in order to “clear” the existing fog and shed a light on the next steps into the future.

## Figures and Tables

**Figure 1 jpm-14-00443-f001:**
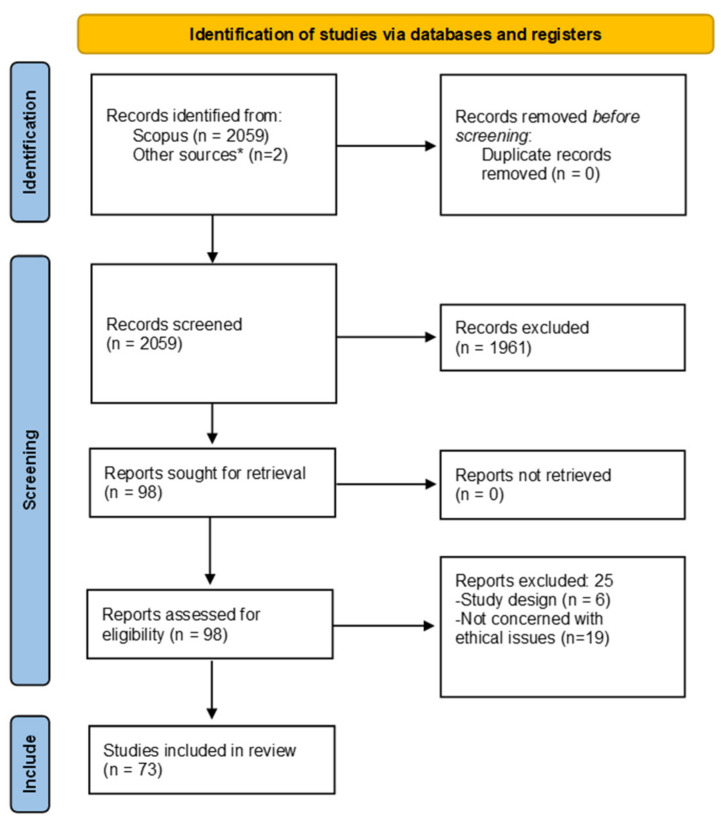
PRISMA flow diagram for study screening and inclusion. * Refers to studies identified from the records retrieved from the database which were not held on the database searched.

**Figure 2 jpm-14-00443-f002:**
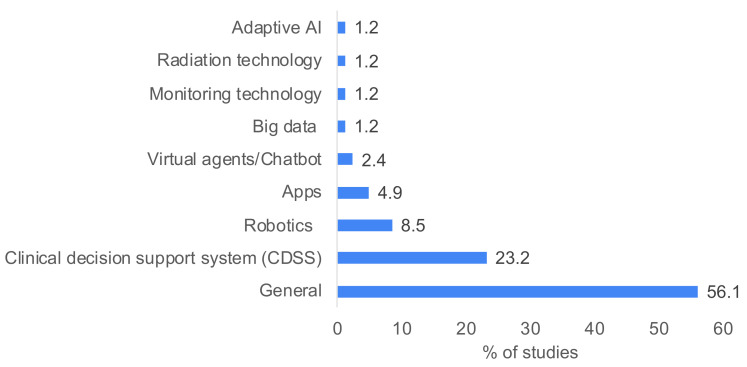
Percentage of studies discussing different technological contexts. Multiple technologies were discussed in some studies.

**Table 1 jpm-14-00443-t001:** Terms/string used for systematic search, divided by area. Each area was then put together with the AND operator. * denotes any character or range of characters in the database search.

Area	String
Ethics terms	Ethic* OR bioethic* OR ((cod*)AND(ethic*)) OR ((ethical OR moral) AND (value* OR principle*)) OR deontolog* OR (meta-ethics) AND (issue or problem or challenge)
Artificial intelligence terms	(“artificial intelligence”) OR (“neural network”) OR (“deep learning”) OR (“deep-learning”) OR (“machine learning”) OR (“machine-learning”) OR AI OR iot OR (“internet of things”) OR (expert system)
Healthcare technology terms	Health OR healthcare OR (health care) OR (medical device*) OR (medical technolog*) OR (medical equipment) OR ((healthcare OR (health care)) AND (technolog*))

**Table 2 jpm-14-00443-t002:** Proportion of studies reporting common ethical themes and the proposed solutions. Others included human bias, value, beneficence, nonmaleficence and integrity.

Ethical Themes	Number of Studies (%)	Proposed Solution/s
Transparency	40 (17)	Multidisciplinary approach to development Standardised framework, regulation and policy Education on AI technology for healthcare professionals and users
Algorithmic bias	30 (13)	
Confidentiality	33 (14)	
Fairness	24 (10)	
Trust	34 (15)	
Autonomy	24 (10)	
Informed consent	13 (6)	
Accountability	25 (11)	
Other	11 (5)	

## Data Availability

The datasets used and/or analysed during this study are available from the corresponding author on reasonable request.

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
