# Peer review of "Clearing the Fog: A Scoping Literature Review on the Ethical Issues Surrounding Artificial Intelligence-Based Medical Devices"

_jpm, 2024, doi:10.3390/jpm14050443_

Round 1

Reviewer 1 Report

Comments and Suggestions for Authors

This is an interesting scoping review of the literature encompassing many underlying ethical issues. Interestingly, the authors try to contextualise the paper in the international environment where numerous documents and guidelines have been published on the related matter.

The manuscript tends to provide a complete overview of ethical, legal and social issues of Medical AI. 

The originality I found in this scoping review it that provides a comprehensive, systematic and up to date overview of the existing literature on ethical, legal and social aspects of medical AI. The complete insight into existing knowledge in the Academic discousrse on the ethical, legal and social aspects of the Medical AI. The conclusions are consistent with main objectives of the proposed study, findings and main arguments.

Author Response

We thank the reviewer for the precious feedback.

Reviewer 2 Report

Comments and Suggestions for Authors

Dear authors,corrections are listed below:

There are no ANDs in the Artificial intelligence terms section in Table 1. This way you should get very uncorrelated results.

In Figure 1, a screenshot was taken with some words underlined. The shape needs to be changed. Additionally, a smaller image can be presented by removing the spaces within the box.

Tables 2 and 3 may be more understandable if presented graphically.

Author Response

We thank the reviewer for the precious feedback.

Table 1 presents the string divided by area. Each area was then put together by the AND operator. This was mentioned in the text and now is also mentioned in the caption.

Figure 1 has been corrected, and the boxes has been reduced in size, maintaining consistency among the boxes sizes.

Table 2 was kept as original with a slight modification, because it contained extra info not possible to be represented in a pie chart.

Table 3 was turned into a pie chart in figure 2.

Reviewer 3 Report

Comments and Suggestions for Authors

This is an interesting work that comes to clarify issues concerning the application of AI in medical devices. It is well structured and technically sound.

Author Response

(The authors gave the same response as above.)
